# A New Gene *SCY3* Homologous to *Scygonadin* Showing Antibacterial Activity and a Potential Role in the Sperm Acrosome Reaction of *Scylla paramamosain*

**DOI:** 10.3390/ijms24065689

**Published:** 2023-03-16

**Authors:** Su Long, Fangyi Chen, Jishan Li, Ying Yang, Ke-Jian Wang

**Affiliations:** 1State Key Laboratory of Marine Environmental Science, College of Ocean & Earth Sciences, Xiamen University, Xiamen 361102, China; 2State-Province Joint Engineering Laboratory of Marine Bioproducts and Technology, College of Ocean & Earth Sciences, Xiamen University, Xiamen 361102, China; 3Fujian Innovation Research Institute for Marine Biological Antimicrobial Peptide Industrial Technology, College of Ocean & Earth Sciences, Xiamen University, Xiamen 361102, China

**Keywords:** *Scylla paramamosain*, antimicrobial peptide, *Scygonadin*, *SCY3*, acrosome reaction

## Abstract

In the study, a new gene homologous to the known antimicrobial peptide *Scygonadin* was identified in mud crab *Scylla paramamosain* and named *SCY3*. The full-length sequences of cDNA and genomic DNA were determined. Similar to *Scygonadin*, *SCY3* was dominantly expressed in the ejaculatory ducts of male crab and the spermatheca of post-mating females at mating. The mRNA expression was significantly up-regulated after stimulation by *Vibrio alginolyticus*, but not by *Staphylococcus aureus*. The recombinant protein rSCY3 had a killing effect on *Micrococcus luteus* and could improve the survival rate of mud crabs infected with *V. alginolyticus*. Further analysis showed that rSCY3 interacted with rSCY1 or rSCY2 using Surface Plasmon Resonance (SPR, a technology for detecting interactions between biomolecules using biosensor chips) and Mammalian Two-Hybrid (M2H, a way of detecting interactions between proteins in vivo). Moreover, the rSCY3 could significantly improve the sperm acrosome reaction (AR) of *S. paramamosain* and the results demonstrated that the binding of rSCY3, rSCY4, and rSCY5 to progesterone was a potential factor affecting the sperm AR by SCYs on. This study lays the foundation for further investigation on the molecular mechanism of SCYs involved in both immunity and physiological effects of *S. paramamosain*.

## 1. Introduction

The complex process of mammalian reproduction is based on a series of highly regulated and synchronized physiological events, and the spermatozoon acrosome reaction (AR) is an exocytotic event critical for the development of mammalian fertilization [1]. Dan was the first to clearly document profound structural changes in the acrosomes of sea urchin spermatozoa prior to fertilization [2]. The acrosome and AR have been studied in a variety of animal species from hydrozoans to humans [3]. In mammals and other vertebrates, the morphological changes and molecular mechanisms of the sperm AR have been intensively studied [4]. Many proteins are considered key players in the AR [1], including CD9, IZUMO1, JUNO, SPACA6, TMEM95, SOF1, ZP, FIMP, DCST1/DCST2, etc. [5]. In marine invertebrates, namely, sea urchins, starfish, clams, oysters, and worms, Bindin, EBR1, suREJ−1/−3, Lysin, sp18, and VERL are the key fertilization proteins [6]. The morphological features of AR have been studied in some crustaceans, such as *Eriocheir sinensis*, where the AR occurs in four major phase: (1) contraction of the radial arms and ejection through an apical aperture; (2) eversion of the acrosomal vesicle; (3) extension of the acrosomal tubule; (4) segregation of membranous lamellar from spermatozoa after AR. In *Penaeus monodon*, the AR undergoes two main stages: (1) acrosomal exocytosis and (2) spheral shape [7]. In *S. paramamosain*, the AR process is divided into three typical steps, as described in [8], namely (1) eversion of acrosomal vesicles; (2) extension of acrosomal tubules and contraction of the nuclear cup; (3) disappearance of acrosomal vesicles and the completion of the reaction. The molecular information during AR is relatively limited in invertebrates: in Chinese mitten crab, p38 MAPK participates in spermatogenesis and the AR [9]; Es-ADAM10 and Es-ADAM17 were also involved in spermatogenesis [10]; Esserpin-3 is an intrinsic sperm protein involved in the regulation of sperm maturation and the AR [11]; UBS27 and UBL40 play key roles in gametogenesis and reproductive success; and ERK was involved in the process of AR of *E. sinensis* [12]. Since the AR-associated molecules (e.g., CatSper) have not been identified in the mud crab, the molecular basis of PG-induced AR of the mud crab remains unclarified, and SCY2 and Screprocin were crucial proteins in AR of *S. paramamosain* [13]. The molecular processes during AR deserves further research in *S. paramamosain*.

*Scylla paramamosain* is very popular all around the world, not only for its delicious taste and high nutritional value, but also for its economic benefits. It is an important aquaculture crab in Southeast Asia [14]. There were six stages in the life cycle of *S. paramamosain*, including embryo, zoea, megalopa, juvenile, sub-adult, and adult [15], in which a crab undergoes multiple molts, with the last molt (reproductive molt) of the female crab initiating the mating process [13]. Successful mating is an important basis for mud crab reproduction. In recent years, with the expansion of crab aquaculture scale, the planning of mud crab aquaculture areas has become unreasonable and unscientific, and the marine environment is complex, resulting in the outbreak of diseases in mud crab farms, causing huge losses annually, and seriously hindering the development of the mud crab aquaculture industry [16].

Antimicrobial peptides (AMPs), one of the most crucial molecules in innate immunity, play an important role in the immune defense of vertebrates and invertebrates [17,18]. AMPs play an immune role in the blood circulation system, and have a protective effect on the reproduction and development of animals [19,20]. It has been reported that there are specific and highly expressed AMPs in the reproductive system of several animals, such as β-defensins from mammalian reproductive system [21]. Andropin was identified in the reproductive tract of male *Drosophila melanogaster* [22,23], and Ceratotoxins have been reported in the accessory glands of sexually mature female Medfly *Ceratitis capitata* [24,25], both of which have strong antibacterial effects and protect sperm during reproduction to ensure the completion of insemination, indicating that they exert a reproductive immunity function in the special evolutionary environment.

As a marine invertebrate, the innate immunity of *S. paramamosain* is the first line of defense against pathogen invasion. AMPs are also key immune-related molecules in marine invertebrates, including crustaceans. Previously, a highly expressed gonadal AMP, named *Scygonadin* in *S. paramamosain*, was originally isolated from the seminal plasma of male mud crabs in 2006 [26], and its homologous genes *SCY2* [14] and *SCY4* and *SCY5* [27] have been identified since. Immunofluorescence detection showed that SCY4 and SCY5 were localized on sperm cells at different developmental stages and in the acrosomal tubes of acrosome reactions, and *SCY5* was found to be up-regulated, while *SCY4* was down-regulated upon stimulation with progesterone in vivo. Scyreprocin [28,29], a protein that interacted with SCY2, was further confirmed to also be a highly expressed AMP in the gonad. It not only has strong antibacterial activity, and participates in reproductive immune process, but has also been shown to be involved in the sperm AR of *S. paramamosain*, which shed light on that AMPs have dual functions of antibacterial and physiological functions. Binding to Ca^2+^ might be the main mechanism by which SCY2 affects the sperm AR of *S. paramamosain*. Therefore, other AMPs have similar functions in the AR of mud crabs, and understanding the relationship between these proteins is worthy of investigation.

In the present study, a new cDNA sequence homologous to *Scygonadin* was cloned and named *SCY3*. Subsequently, the expression profiles of *SCY3* at different developmental stages and in various tissues of adult *S. paramamosain* were investigated. The mRNA expression pattern of *SCY3* in the hepatopancreas of crabs challenged with *Vibrio alginolyticus* or *Staphylococcus aureus* was analyzed. The recombinant protein (rSCY3) was obtained in the *Pichia pastoris* eukaryotic expression system, and the antimicrobial activity in vitro and anti-infective activity in vivo was detected. In addition, the interactions among the five SCYs proteins were investigated by Surface Plasmon Resonance (SPR) and Mammalian Two-Hybrid (M2H). The effect of SCY3, SCY4, and SCY5 proteins treatment on the AR of sperm was analyzed. Additionally, the potential reason why SCYs affect sperm AR was studied by detecting the affinity between the SCY proteins and progesterones.

## 2. Results

### 2.1. Gene Cloning and Sequence Analysis of SCY3

The full-length cDNA sequence *SCY3* (GenBank accession No. MZ422779) was obtained by RACE PCR analysis. The complete sequence of 887 bp contains a 372 bp open reading frame, encoding a 124 amino acid protein, with 63.7%, 61.3%, 51.6%, and 54.6% homology to *SCY1* [26], *SCY2* [30], *SCY4*, and *SCY5* [27], respectively. It contains 69 bp of 5′ untranslated region (UTR), and 446 bp of 3′ UTR with a poly(A) tail (Figure 1A,B), and the molecular mass of the mature peptide is 10.8 kDa. The DNA sequence of *SCY3* is 1692 bp (GenBank accession No. MZ448189), and its structure is the same as *SCY1* and *SCY2*, all of which containing three exons and two introns (Figure 1C,D). The tertiary structure of *SCY3* predicted by PHYRE2 was composed of three α-helices and three β-sheets (Appendix A).

### 2.2. The Tissue Distribution and Immune Response of SCY3 mRNA in Hepatopancreas Stimulated by V. alginolyticus and S. aureus

Absolute quantitative PCR (qPCR) was used to detect the expression of *SCY3* mRNA in embryos, zoea larval stage, megalopa larval stage, juvenile, and tissues of adult male and female crabs. The highest expression level of *SCY3* was found to be at the embryonic stage on day 5 after hatching (Figure 2A). In adult crabs, *SCY3* was widely distributed in different tissues, similar to *SCY1* and *SCY2* of the same family, and all of them were predominantly expressed in the ejaculatory ducts, up to about 5 × 10^8^ copies (Figure 2B,C), slightly higher than *SCY1* (6 × 10^7^ copies) [31], but lower than *SCY2* (3 × 10^9^ copies) [14].

Relative qPCR results showed that the transcripts of *SCY3* gene was significantly up-regulated at 6 h in hepatopancreas under *V. alginolyticus* challenge (Figure 2D), but not obviously changed after infection with *S. aureus* (Figure 2E). *SCY3* mRNA transcripts in the reproductive system of pre- and post-mating female mud crabs were evaluated using qPCR. The results showed that *SCY3* mRNA was remarkably increased in the spermatheca of post-mating females (Figure 2F).

### 2.3. Expression and Purification of rSCY3 in Pichia pastoris GS115 or E. coli

After optimization, 0.5% methanol at pH 6.0 for 24 h was selected as optimal induction condition for rSCY3. Recombinant SCY3 (rSCY3) was successfully obtained in eukaryotic expression system of *P. pastoris*, and the molecular weight of rSCY3 was consistent with the predicted one (10.8 kDa) (Appendix A). The mass spectrometry result confirmed that the amino acid sequence of the purified protein was the target rSCY3 (Appendix A).

In addition, constructed expression plasmid *SCY3*-pET32a in *E. coli* Rosetta (DE3) induced by 0.5 mM IPTG at 28 °C for 8 h was the optimal induction condition. rSCY3-rTrx were successfully obtained in the prokaryotic expression system, and a main single protein band around 25 kDa was observed in Appendix A.

### 2.4. Antimicrobial Activity and Preliminary Antibacterial Mechanism of rSCY3

The antimicrobial activities of the recombinant SCY3 using the MIC assay (Table 1 and Appendix A). The results showed that both prokaryotic and eukaryotic expressed protein SCY3 (rSCY3) had good resistance to *M. luteus* (with MIC value of 24–48 µM). Considering that the similar antibacterial activity of recombinant SCY3 obtained by eukaryotic and prokaryotic expression system, rSCY3 proteins were obtained by eukaryotic expression for subsequent analysis.

After co-incubating rSCY3 with *M. luteus*, the surface morphology of the cells was observed by scanning electron microscope (SEM). The results showed that the cell surface of the control group was intact without obvious disruption (Figure 3). In the experimental group, the morphology of *M. luteus* changed significantly, where the cell surface was depressed, ruptured, and showed a significant disruption of membrane integrity.

### 2.5. rSCY3 Increase the Survival of S. paramamosain Infected V. alginolyticus

To determine whether rSCY3 could be safely used in in vivo experiments, the cytotoxicity of rSCY3 was analyzed using hemocytes isolated from healthy male crabs (300 ± 30 g). The results showed an upward but not significant trend in cell viability, indicating that rSCY3 had no cytotoxicity (Figure 4A).

To assess the effect of rSCY3 in vivo, *S. paramamosain* was challenged with *V. alginolyticus*. One hour after bacterial injection, 20 µL of rSCY3 (60 µg) or crab saline was injected into the crabs. The survival rate of the control group decreased to 10 % at 36 h post-injection, while the rSCY3 treatment group showed a 50 % survival rate (Figure 4B). In total, 110 h after challenge, 90 % of the crabs in the control group died, while the survival rate of the experimental group remained around 35 % (Figure 4B). In general, *S. paramamosain* treated with rSCY3 after bacterial challenge exhibited a significantly higher survival rate than that of crabs injected with crab saline until 110 h (*p* = 0.0213).

### 2.6. The Interaction between SCY1-SCY5

Thus far, a total of five SCYs have been found in *S. paramamosain* [14,26,27], a conjecture that the complexes of *scygonadin* homologous protein formed play a similar role in sperm AR raised from the previous study of SCY2 and scyreprocin. Whether these proteins interact with each other to form complexes involved in the activities of mud crab deserves investigation. Recombinant His-tagged SCY1-SCY4 (rSCY1-rSCY4), Trx-tagged SCY5 (rSCY5), and Trx (rTrx) were expressed and purified (Appendix A). Surface plasmon resonance assay (SPR) indicated that the calculated equilibrium dissociation constant (K_D_) of the rSCY1–rSCY2, rSCY1–rSCY3, rSCY1–rSCY4, rSCY1–rSCY5, rSCY2–rSCY3, rSCY2–rSCY4, and rSCY2–rSCY5 interactions was 11.26 µM, 4.12 µM, 6.86 µM, 0.14 µM, 0.62 µM, 14.02 µM, and 0.15 µM (Figure 5 and Table 2), respectively, and the interaction was further verified by a M2H assay (Figure 6). There is no interaction for rSCY3–rSCY4, rSCY3–rSCY5, and rSCY4–rSCY5 (Appendix A).

### 2.7. Co-incubation of rSCY3 Combination with Sperm Significantly Improved the AR

Whereas previous studies have shown that SCY2 protein has a potential function in AR, SCY4 and SCY5 were localized on sperm cells at different developmental stages and in the acrosomal tubes of acrosomal reactions. To detect the effect of recombinant proteins SCYs (rSCY3, rSCY4, and rSCY5) on sperm AR. The induced AR rate in the rSCY5 treatment groups were about 66.90 ± 3.39%, which were significantly high than that in the control group (50.3 ± 2.40%, *p* = 0.0117). Compared with the control, the AR rates of sperm treated with rSCY3 and rSCY4 were 92.10 ± 0.85% and 80.55 ± 1.34%, respectively, which significantly improved the AR (*p* = 0.0013 and 0.0361) (Figure 7).

### 2.8. rSCYs Binding to Progesterone Might Be a Potential Mechanism Affecting the Sperm AR

In order to further explore how rSCYs affect AR-induced progesterone (P4) in the sperm of the mud crab, we performed a biolayer interferometry assay (BLI) to further determine the interaction of P4 with rSCY3, rSCY4, and rSCY5. The results showed that their KD values were 3.08 × 10^−2^ M, 2.28 × 10^−3^ M, and 2.53 × 10^−2^ M, respectively. Therefore, the affinity of progesterone and rSCY3-rSCY5 was rSCY4 > rSCY5 > rSCY3 (Figure 8A–C). Binding to progesterone might be the potential reason why rSCYs affected the sperm AR of the mud crab.

## 3. Discussion

The complex marine environment makes the reproductive process of marine organisms more susceptible to microorganisms than terrestrial organisms [32]. We aim to identify and utilize AMPs that are highly expressed in the reproductive system of *S. paramamosain* [14,16,27,28,29,30,31,33]. Recent studies have shown that SCY2 and its interacting protein Scyreprocin not only have antibacterial activity, but also participate in the AR as a crucial factor in the fine and complex AR of mud crabs [13]. Are there other AMPs with similar functions? We devoted to identify more AMPs in the reproductive system to study their antibacterial activities and physiological functions. In the present study, a new homologous *Scygonadin* gene, named *SCY3*, was identified in *S. paramamosain*. It had similar structure and expression characteristics to known SCY genes. Functional analysis showed that the recombinant protein rSCY3 could kill *M. luteus*. Furthermore, rSCY3 could significantly improve the survival rate of *S. paramamosain* infected with *V. alginolyticus* in vivo. The results showed that rSCY3 could interact with rSCY1 or rSCY2. Additionally, the rSCY3, rSCY4, and rSCY5 can significantly increase the ratio of AR in vitro, and the affinity of three rSCYs to progesterone is the potential reason why SCYs affect the sperm AR in *S. paramamosain*.

The AMP families have been identified using their unique characteristic, with defensins being widely distributed in chicken, pigs, dog, chimp, in mice, humans, and so on, and with abundant 3–4 kDa AMPs that are variable cationic and contain six disulfide-paired cysteines [34,35]. In crustaceans, the anti-lipopolysaccharide factors (ALFs) encode a signal peptide (22 to 28 residues) and a mature peptide (10.74 to 12.23 kDa), containing two conserved cysteine residues, of which the isoelectric points (pI) varies between 5.0 and 11.0. The three-dimensional structure including three α-helices packed with four β-sheets [33,36]. The crustins usually have a signal peptide with a length of 16–24 amino acids (aa) at the N-terminus, and a hydrophobic core consisting of eight cysteine residues forming four disulfide bonds at the C-terminus, also known as disulfide cores (DSC) [37]. There is some similarity among *SCY1*, *SCY2*, *SCY3*, *SCY4*, and *SCY5*, which encode a precursor composed of a signal peptide (24 aa), followed by a mature peptide (MW 73 to 102 kDa), with a pI below 7.0 (4.9–6.9), belonging to the anionic protein [38]. The genomic structures of *SCY1* [26], *SCY2* [14], *SCY4* [27], and *SCY3* contain three exons and two introns, except for SCY5 (which contains two exons and one intron). Based on the full-length amino acid sequence alignment, the homology between *SCY3* and *SCY1*, *SCY2*, *SCY4*, and *SCY5* was 51.6–63.7%, and the homology between SCYs was about 50%. We grouped the five known SCYs into the SCYs AMP family due to homology and basic sequence properties.

β-defensin was specifically expressed in the male epididymis [39,40,41,42,43,44,45,46,47,48] and female reproductive tract [19,20]. Even though these AMPs are specifically expressed in the reproductive system, they have different response patterns when affected by hormone stimulation [49]. Similarly, SCYs in *S. paramamosain* were highly expressed in the ejaculatory duct of male crabs and may transfer to the spermatheca of female crabs along with semen during mating [14,27], as did *SCY3* in this study. In addition, compared with other SCY genes, *SCY1*, *SCY2*, *SCY4*, and *SCY5* were not affected by *V. alginolyticus* or LPS stimulation, while the expression of *SCY1*, *SCY2*, and *SCY5* was up-regulated and the expression of *SCY4* was down-regulated when stimulated by progesterone. In this study, *SCY3* was significantly up-regulated in hepatopancreas at 6 h after *V. alginolyticus* stimulation, and extremely significantly up-regulated in hemocyte at 24 h after *S. aureus* stimulation (Appendix A), while the ejaculation duct showed no significant change after progesterone stimulation (Appendix A). This may indicate that the SCYs family genes had different responses to co-maintain the stability of the internal environment of *S. paramamosain* under bacterial or progesterone stimulation.

From protein synthesis, maturation, and degradation to vesicle budding, trafficking, and fusion and from receptor dimerization, signaling cascades, and gene regulation to metabolism and catabolism, almost all cellular functions depend on complex protein–protein interactions (PPIs), and are executed by it [50,51]. The delicate and complex reproductive processes of living organisms, including the acrosomal reaction, had been extensively studied on PPIs between homologous protein. The seminal vesicle secretion (SVS), SVS1–SVS3, cross-linked each other through transglutaminase to form a copulatory plug [52]. SVS3 showed affinity for SVS2 and facilitated the effects of SVS2 on sperm capacitation [53]. Epididymal protease inhibitor (EPPIN) directly binds SVS2, while indirectly interacting with SVS3A, SVS5, and SVS6 via SVS2 in mice [54]. As the most biochemically complex ion channel known to date, the CatSper channel complex consist of seven protein components (CatSper1-4, β, γ, and δ) derived from mouse testis [55,56]. SCY2 can interact with Scyreprocin as crucial molecules involved in the sperm AR of mud crabs [13]. In this study, the interaction between SCY1–SCY5 was analyzed by M2H assay; the recombinant proteins rSCY1–rSCY4 were obtained in eukaryotic expression system, and the affinity was determined by SPR. The results indicated that rSCY1 interacted rSCY2, rSCY3, rSCY4, and rSCY5; rSCY2 interacted with rSCY3, rSCY4, and rSCY5. These results suggested that SCY proteins may be involved in immune and reproductive functions in *S. paramamosain*. However, it is unclear what complexes these proteins form, such as a CatSper channel, SVS proteins, or other new types, and the mechanisms by which they work. Those are the questions we would like to further clarify.

Progesterone, a hormone involved in AR [57,58], has been used as an inducer in studies of AR affected by related proteins in vitro [13]. After incubation of the protein, the motility, capacitation, and AR of sperm are commonly used to evaluate the function of the protein in rodents. The porcine myeloid AMP-37 (PMAP-37) seems to be a suitable candidate to replace antibiotics in extended semen, as it hardly impairs sperm viability in Pietrain boars [59]. The addition of AMP protegrine 1 (PG1) to the semen extender was effective in improving sperm viability in boars [60]. Recombinant SPINK3 (Serine Protease Inhibitor Kazal-type 3) improves sperm quality and in vitro fertility in rams [61]. Spink3 modulated the mammalian sperm activity, and affected AR in mice [62,63]. In invertebrates, the AR was also the indicator of detecting the effect of protein on sperm [11,12,13]. In this study, the AR of sperm treated with rSCY1-rSCY5 was detected by flow cytometry, and it was found that the ratio of AR varied with different treatments. The rSCY3, rSCY4, and rSCY5 significantly increased the induced AR ratio, as well rSCY2, indicating that the SCYs could promote the sperm AR. In addition, the BLI assay is emerging as a novel label-free methodology for detecting protein–small molecule interactions by immobilizing a tiny amount of protein sample on the surfaces of biosensor and measuring the optical changing signals [64]. Additionally, the analysis of recombinant human erythropoietin (rh-Epo)/liposome interactions [65] showed that small molecule inhibitors (SMI) bound to Tissue transglutaminase (TG2) potently inhibited cancer cell adhesion [66]. In the present study, we obtained the binding kinetics of rSCYs proteins/progesterone using BLI instead of ELISA, as used in our previous research [13], to show that there is a binding effect between rSCYs proteins and progesterone, which was concentration dependent. Preliminary studies suggested that rSCYs proteins may affect AR by binding to progesterone. However, the deeper molecular mechanism underlying the effects of SCYs on AR deserved further investigation.

Sperm capacitation refers to the entry of sperm into the female reproductive system after stimulation with substances secreted by the female, resulting in a series of changes in the surface of the sperm plasma membrane, including transfer of components and rearrangement of particles in the membrane, as well as the activation of acrosomal enzymes in the sperm, before the sperm has the ability to fertilize [8], a process that has been demonstrated in mammals [67] and even in crayfish [7], while it is indistinguishable in *S. paramamosain*. In previous studies, when the induced acrosome reaction of sperm obtained from males and females were treated under the same conditions, the acrosome reaction of sperm obtained from males could not be fully completed. In general, sperm respond only to the contraction of the radial arm without the protrusion of the head cap [8], and the study conducted by our team also found that the % AR of sperm collected from female spermathecae was significantly higher than those from males [13]. Based on these results, it is possible that the sperm of *S. paramamosain* has undergone a capacitation process during the storage of sperm after mating, allowing the morphologically mature sperm to reach physiological maturity with the ability to complete fertilization of the acrosome reaction. However, further understanding of the morphological and molecular changes during sperm capacitation is required. In the present study, we mainly focused on the expression and distribution characteristics of *SCY3* and its functions as a new Scygonadin-like antimicrobial peptide for comparison with the reported SCY1, SCY2, SCY4, and SCY5, and the potential function of the AR based on the previous study that SCY2 and Screprocin play a dual role in both reproductive immunity and PG-induced AR of mud crab *S. paramamosain*. Whether SCY3 plays an important role in sperm capacitation is worth further study.

## 4. Materials and Methods

### 4.1. Animals and Reagents

About 250 *S. paramamosain* with a weight of about 300 ± 30 g were purchased from a crab farm in Xiamen for tissue sample collection. In total, 40 crabs weighing about 60 ± 10 g were used for immunoprotection experiments. *E. coli* DH5α was used for gene subcloning. Moreover, pET32a, *E. coli* BL21(DE3) plysS strain and *E. coli* Rosetta (DE3) purchased from TransGen (TransGen Biotech,Beijing, China) were used for the prokaryotic expression system. The standard bacterial strains were purchased from the Institute of Microbiology, Chinese Academy of Sciences, and listed in Appendix A. The restriction enzymes EcoR I, Not I, Kpn I, and Xba I were purchased from Thermo Fisher (Thermo Fisher Scientific, Waltham, MA, USA).

### 4.2. Full-Length cDNA and DNA Sequences Cloning of SCY3

Total RNA of normal crab ejaculation ducts was prepared using Trizol reagent (Invitrogen, Waltham, MA, USA) following the manufacturer′s instructions. According to the partial cDNA sequence of *SCY3* from the transcriptome database established by our laboratory, primers (shown in Appendix A) were designed through PrimerPrimer5 software (Primer, Ottawa, ON, Canada). The cDNA was synthesized following the instructions of SMARTer^®^ RACE 5′/3′Kit User Manual (Takara, Dalian, China), RACE PCR and the expected fragments were performed according to the standard operations. The obtained sequences were then subjected to blast and cluster analysis for verification.

TIANamp Marine Animal Tissue Genomic DNA Extraction Kit (Tiangen Biotech, Beijing, China) was used to obtain genomic DNA of male crab ejaculation ducts. PCR was performed using primers (Appendix A) and the target fragments were then purified and sequenced.

### 4.3. Sequence and Phylogenetic Analysis

Homology analysis of nucleotides and deduced amino acids was performed using DNAMAN. Signal peptide (SP) was identified using SignalP5.0 program (http://www.cbs.dtu.dk/services/ SignalP) at 24 September 2019. Domain and motif analysis was predicted using the conserved domain search program of NCBI (https://www.ncbi.nlm.nih.gov/) and the Smart database (http://smart.embl-heidelberg.de/s mart/) at 6 November 2019. The theoretical molecular mass and isoelectric point were predicted using ExPASy (http://web.expasy.org/protparam) at 31 October 2019. The PSIPRED Workbench (http://bioinf.cs.ucl.ac.uk/psipred/) was used to predict the secondary structure at 14 March 2020. The tertiary structure was predicted by PHYRE2 (http://www.sbg.bio. ic.ac.uk/phyre2/html/) at 8 April 2020.

### 4.4. The Expression Pattern of SCY3 in Different Developmental Stages and Various Tissues of S. paramamosain

Hemolymph of normal adult mud crabs, including males (300 ± 30 g, n = 5) and females (200 ± 30 g, n = 5), were collected [14]. Crab samples at different developmental stages were collected by our laboratory. Other tissues, including eyestalk, gills, heart, mid-gut gland, hepatopancreas, muscle, stomach, subcuticular epidermis, nerves, and tissues from the reproductive systems of pre- or post-mating female crabs (spermatheca, reproductive duct, and ovaries) and male crabs (testes, anterior vas deferens, seminal vesicle, posterior vas deferens, ejaculation ducts, and posterior ejaculation ducts), were also sampled and ready for total RNA isolation and cDNA synthesis according to the standard protocol. Absolute and relative quantitative real-time PCR (qPCR) using primers (Appendix A) were performed following the regular program, and the absolute quantification measures the number of copies in 1 µg of RNA. qPCR soft 3.4 (Analytik Jena, Jena, Germany) was used to calculate the original data. For relative quantitative real-time PCR, the GAPDH gene was employed as an internal standard, and the fold change of gene expression relative to the control was determined by the 2−ΔΔCt method.

### 4.5. The Immune Challenges of V. alginolyticus and S. aureus to Crabs

For the immune challenge experiments, healthy male *S. paramamosain* (300 ± 30 g) were purchased from a commercial crab farm in Xiamen, China, and acclimated at 25 ± 1 °C for 3–4 days. The challenge experiments were conducted as described previously [33]. Hepatopancreas were collected from five crabs per group at 0, 3, 6, 12, 24, 48, 72, and 96 h post-injection (hpi). Tissues were packed by sterilized tin foil and then stored in −80 °C for later use. Total RNA was extracted, followed by relative qPCR using gene-specific primers (as listed in Appendix A).

### 4.6. Optimization and Purification of Recombinant SCY3 (rSCY3) in the Eukaryotic and Prokaryotic Expression System

The recombinant expression plasmid of *SCY3* mature peptide was constructed using the primers listed in Appendix A. After 24 h of induction with 0.5% methamphetamine for large-scale expression, the target protein was separated from the yeast precipitate, dialyzed against phosphate buffer, and purified by Ni^2+^ affinity chromatography. The purity of the protein was analyzed by SDS-PAGE combined with Coomassie brilliant blue staining, and the concentration was determined by a Bradford protein assay kit (Beyotime Institute of Biotechnology, Shanghai, China). The recombinant protein, with a purity of more than 90%, was frozen and stored at −80 °C for later use. It was further confirmed by the Mass Spectrometry Center, School of Life Sciences, Xiamen University.

In addition, the recombinant expression vector of SCY3 mature peptide was also constructed using pET-32a (the primers used were shown in Appendix A). The recombinant plasmid pET-32a-*SCY3* was transformed into *E. coli* Rosetta (DE3) and *E. coli* BL21(DE3) plysS strain. Thioredoxin (Trx) with 6× His-tag was expressed as a negative control. Expression and purification of recombinant proteins SCY3 and Trx in prokaryotic expression system of *E. coli* were processed as previously described [27].

### 4.7. Antimicrobial Assays and Scanning Electron Microscope Examinations of Microbial Morphology

The antimicrobial activity of rSCY3 was analyzed. Microorganisms included nine Gram-negative bacteria, six Gram-positive bacteria, and eight fungi, as listed in Appendix A. The MIC and MBC values were determined in accordance with the method of liquid growth inhibition assay described before [28], which were performed three times independently.

For scanning electron microscope (SEM) observation, a sterile tube containing only 1 mL of growth medium and 1 mL of growth medium supplemented with rSCY3 was inoculated with approximately 5 × 10^5^ CFU of microbes. After incubation for 60 min, microbial cells were collected and fixed with pre-cooled 2.5% (*w*/*v*) glutaraldehyde for 2 h at 4 °C. Microbes were dehydrated and gold-coated as previously described prior to observation by a ZEISS SUPRA^TM^ 55 scanning electron microscope (ZEISS, Oberkochen, Germany).

### 4.8. Cytotoxicity Assay

Hemocytes from healthy male crab (200 ± 10 g) were maintained in L15 medium supplemented with 5% FBS and 1.2% NaCl. Cells were seeded at ~2.0 × 10^4^ cells well^−1^ in a 96-well cell culture plate (Thermo Fisher, Waltham, MA, USA), and cultured at 26 °C without CO_2_ overnight. Hemocytes were incubated with rSCY3 (0, 1, 2, 4, 8, 16, 32, and 24 μM). After 24 h of incubation, cell viability was assessed using the CellTiter 96^®^AQueous Kit (Promega, Madison, WI, USA). The experiments were carried out in triplicate.

### 4.9. Effects of rSCY3 on the Survival Rate of S. paramamosain

In total, 40 mud crabs (60 ± 10 g) were acclimated for about one week before being subjected to the experiment. *V. alginolyticus* was cultured to the logarithmic phase and washed three times. A total of 20 crabs were randomly selected, and each crab was injected with 25 μL *V. alginolyticus* (5 × 10^7^ CFU) with a micro-injector. One hour after the bacterial injection, 20 μL (about 3 μg/µL) of rSCY3 were injected into these crabs, and a group injected with an equal volume of crab saline (NaCl 496 mM, KCl 9.52 mM, MgSO_4_ 12.8 mM, CaCl_2_ 16.2 mM, MgCl_2_ 0.84 mM, NaHCO_3_ 5.95 mM, HEPES 20 mM; PH = 7.4) was used as control. The number of dead crabs at different time points (0, 3, 6, 9, 12, 24, 36, 48, 72, 96, 118 h) was recorded, and a survival curve was drawn.

### 4.10. Preparation of Recombinant Proteins and SPR Assay

High-purity recombinant proteins rSCY1 [68], rSCY2 [14], rSCY3 (as shown above), and rSCY4 [27] were obtained by the *pichia pastoris* eukaryotic expression system. The recombinant protein rSCY5 and rTrx were obtained by prokaryotic expression system [27].

Surface plasmon resonance (SPR), a technology for detecting interactions between biomolecules using biosensor chips. All experiments below were carried out using a BiacoreT200 and CM5 sensor chip (GE Healthcare, Pittsburgh, PA, USA). To evaluate the interaction between SCY1-SCY5, rSCY1–4 was immobilized on flow cell 2 or 4 (flow cell 1 and 3 were used as a blank or rTrx control) of the CM5 sensor chip to a total signal increase of ∼200 resonance units. After pH scouting, pre-immobilization, and the single cycle kinetics, a multiple cycle kinetic assay was performed in running PBS (28.8, 14.4, 7.2, 3.6, 1.8, 0.9, and 0.45 µM) at a flow rate of 30 mL min^−1^ at 25 °C. After baseline equilibration, rSCY1–5 were diluted in PBS, injected, and analyzed. Data were recorded at a rate of 10 Hz during 120-s association and 120-s dissociation phases.

### 4.11. M2H Assay

The mammalian two-hybrid (M2H) assay, a way of detecting interactions between proteins in vivo, was performed on human cervical carcinoma cell (HeLa cells, kindly provided by Stem Cell Bank, Chinese Academy of Sciences) and human embryonic kidney 293T cells (HEK 293T cells) using the Checkmate^TM^ M2H system (Promega, Madison, WI, USA) in accordance with the manufacturer’s instructions. Briefly, the gene fragments encoding SCYs mature peptide were cloned in frame with pACT vector and pBIND vector (primer sequences were listed in Appendix A), respectively. HeLa cells and HEK293T cells were maintained in Dulbecco’s modified Eagle’s medium (Invitrogen, Waltham, MA, USA) supplemented with 10% fetal bovine serum (FBS; Gibco, Waltham, MA, USA). Cells were plated at ∼3.0 × 10^5^ cells well^−1^ on a 96-well cell culture plate (Thermo Fisher Scientific, Waltham, MA, USA). Vector combinations, including (1) pACT/pBIND/pG5luc; (2) pACT-A/pBIND/pG5luc; (3) pACT/pBIND-B/pG5luc; (4) pACT-MyoD/pBIND-Id/pG5luc; (5) Blank; and (6) pACT-A /pBIND-B/pG5luc, were transiently transfected into HeLa cells and HEK293T cells using Lip3000 (Invitrogen, Waltham, MA, USA) according to the manufacturer’s instructions. Cells were harvested at 24 h and 48 h post-transfection and lysed in passive lysis buffer (Promega, Madison, WI, USA). Reporter gene activities were measured using the Dual-Luciferase Reporter Assay System (Promega, Madison, WI, USA) on a GloMax 20/20 luminometer (Promega, Madison, WI, USA), the background level of firefly luciferase expression was obtained from the pG5luc vector, *Renilla* luciferase activity was determined from the pBIND control vector or pBIND-X vector, and the luminescence (RLU) was the ratio of *Renilla* luciferase and firefly luciferase. The experiments were performed in five replicates, and repeated twice.

### 4.12. Measuring the AR Responded by the rSCY3-rSCY5

Spermatozoa were obtained from the mixed spermatophores of five post-mating female crabs and placed in pre-cooled (4 °C) Ca^2+^-free artificial seawater (Ca^2+^-FASW) (Tris, 4.8440 g; NaCl, 55.5180 g; KCl, 1.7892 g; MgCl_2_⋅6H_2_O, 12.1980 g; EGTA, 0.7608 g 2 L, pH 8.2). First, the spermatophores were extruded slightly from the spermatheca with forceps in Ca^2+^-FASW with 1% 100 × Penicillin Streptomycin solution (Thermo Fisher Scientific, Waltham, MA, USA), and then they were cut up with scissors and mixed on a rotating incubator. After 10 min, the mixture was passed through a 100 μm cell strainer (Biosharp, Beijing, China), and centrifuged at 2000× *g* for 15 min. The spermatophores were precipitated, resuspended, and then digested with 0.025% trypsin (GIBCO, Waltham, MA, USA) at 26 °C with gentle stirring for 15 min, after which the reaction was stopped with 10% FBS diluted in Ca^2+^-FASW. The spermatozoal pellet was washed three times with pre-cooled (4 °C) Ca^2+^-FASW to obtain pure sperm cells. The cells were mixed in Ca^2+^- FASW containing 4,6-diamidine-2-phenylidole-dihydrochlo-ride DAPI (ZLI-9557, ZSGB-BIO; Beijing, China) at 26 °C for 15 min, and observed under a DAPI field microscope at ×100 magnification (Leica, Heidelberg, Germany) to confirm the integrity of the cells. The density of the sperm cell suspension was then adjusted to form a suspension with a concentration of 5 × 10^6^ sperm cells/mL, and stored at 4 °C within 24 h for further experiments.

This paper had the following divided groups: (1) ASW (Artificial sea water, ASW, Tris, 2.4220 g; NaCl, 27.7500 g; KCL, 0.8940 g; MgCl_2_⋅6H_2_O, 6.1 g; CaCl_2_, 1.3320, 1 L, pH 8.2) + a single rSCY; (2) FASW (Free Ca^2+^ artificial sea water, FASW)+ a single rSCY; (3) ASW+ a single rSCY + progesterone; (4) FASW+ a single rSCY + progesterone, the concentration of “a single rSCY” was 24 µM, in addition, the groups without protein or the rTrx-added (24 µM) group were set as the control. All groups were treated as previously described [13]. The SYTO9 (Thermo Fisher Scientific, Waltham, MA, USA) was added to the fixed sperm according to the instruction, and the cells were incubated at room temperature for 30 min in the dark, washed twice with FASW by centrifugation (2000 g, 15 min). Flow cytometry data were collected by CytoFLEX S (Beckman Coulter Inc., Brea, CA, USA) and analyzed by FlowJo VX10.0. (National Institutes of Health, Bethesda, MD, USA). The experiment was carried out in triplicate.

### 4.13. Affinity Measurements between Progesterone and rSCY3-rSCY5 by Fortebio’s Octet System

Purified proteins were diluted or dialyzed in dialysis buffer (KH_2_PO_4_ 42 mM, Na_2_HPO_4_ 8 mM, NaCl 136 mM, KCl 2.6 mM, pH 7.4, Tween 20 0.02%) and biotinylated using the EZ-Link^®^NHS-PEO_4_-Biotinylation Kit (Thermo Fisher Scientific, Waltham, MA, USA) for 30 min at room temperature. The unconjugated biotin was removed by Zeba Spin desalting column (Thermo Fisher Scientific, Waltham, MA, USA). The biotin-conjugated protein was diluted to 50 µg/mL. The Super Streptavidin sensors (Fortebio, Silicon Valley, CA, USA) were pre-wetted in dialysis buffer for 15 min prior to use and then loaded with biotinylated proteins for 15 min. The sensors unloaded with biotinylated proteins or loaded with rTrx was used as controls to correct for base-line drift. Progesterone was prepared in serial dilutions (0.125, 0.25, 0.5, 1, 2, and 4 mM) in a 96-well plate. Measurements were carried out automatically at room temperature using the Fortebio’s Octet System and analyzed by Octet System Data Analysis. Every experiment was repeated twice.

### 4.14. Statistical Analysis

Statistical analysis was performed using IBM SPSS statistics (version 22; IBM Corp., Armonk, NY, USA) and GraphPad Prism 9.0 Software (GraphPad Software Inc., California, CA, USA), with a confidence level of 95% being considered statistically significant. Data were shown as mean ± standard deviation.

## 5. Conclusions

In the present study, a new gene homologous to *Scygonadin* was identified from the mud crab *S. paramamosain*, named *SCY3*. The full length of cDNA and DNA sequences were determined. The mRNA expression analysis showed that *SCY3* was dominantly expressed in the ejaculatory duct of male crabs, and was also detected in the spermatheca of post-mating female mud crabs. In addition, *V. alginolyticus*, but not *S. aureus*, could significantly up-regulated the expression of *SCY3* gene in the hepatopancreas. The recombinant protein rSCY3 had antibacterial activity against *M. luteus*, and significantly improved the survival rate of mud crabs infected with *V. alginolyticus*. These results suggested that SCY3 possessed similar expression characteristics and antibacterial activity to SCY family genes. Further studies showed that rSCY3 interacted with rSCY1 or rSCY2. Additionally, the rSCY3, rSCY4, and rSCY5 can significantly improve the AR of sperm, and the affinity of rSCYs to progesterone might be a potential factor that SCYs affect the sperm AR. Taken together, our study demonstrated the expression characteristics and antibacterial activity of SCY3, a new member of the SCY family, and revealed that five SCY proteins may be involved in the sperm AR as a complex, and the combination of SCYs and progesterone affected the AR, which provides a new idea for further research on the molecular mechanism of physiological functions of SCYs in mud crabs.

## Figures and Tables

**Figure 1 ijms-24-05689-f001:**
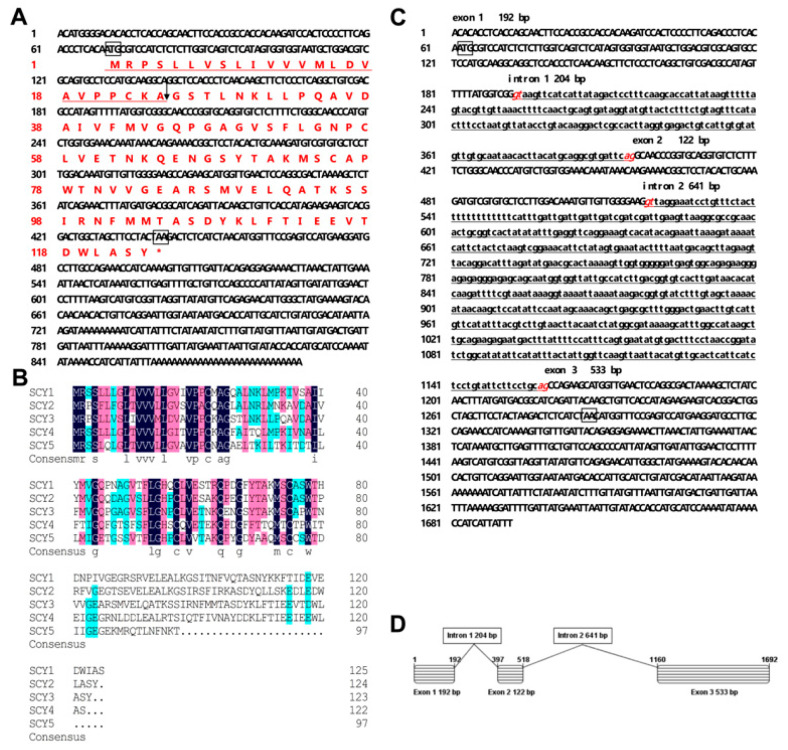
The full-length cDNA sequence, multiple alignment of *SCY3* amino acid sequences with other known SCYs in the mud crab *S. paramamosain*, the full-length DNA sequence, and organization of *SCY3*. (**A**) The full-length cDNA sequences (above) of *SCY3* and its predicted amino acid sequences (below). The arrow indicated the predicted cleavage site of the signal peptide, the proposed start codon and stop codon were boxed, and the red represents the predicted amino acid. (**B**) Multiple alignment of *SCY3* amino acid sequences with other SCYs. The color scheme in (**B**) represents the conservation of the amino acids, and the complete consistency is dark blue. (**C**) The full-length DNA sequences of *SCY3* the intron dinucleotide acceptor and donor sites (gt/ag) for RNA splicing are italics and red, and the start codons and stop codons are boxed. (**D**) Gene organizations of *SCY3*.

**Figure 2 ijms-24-05689-f002:**
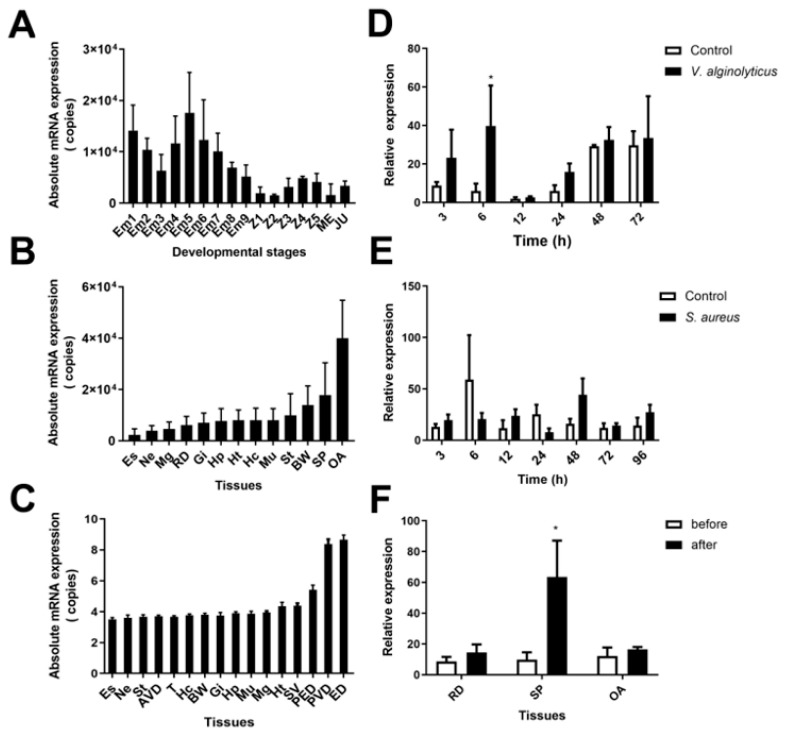
The distribution of *SCY3* in different developmental stages and adult crab tissues, the expression pattern of *SCY3* in hepatopancreas after challenge with *V. alginolyticus* and *S. aureus*, and the mRNA expression in the pre- and post-mating reproductive system of female *S. paramamosain*. (**A**) The expression of *SCY3* at different developmental stages of *S. paramamosain*. Em1–9: the mud crab embryos from day 1 to day 9 post hatching; Z1–5: zoea larval stage 1–5; ME: megalopa larval stage; JU: juvenile. (**B**,**C**) The expression of *SCY3* in various tissues of female crabs (**B**) or male crab (**C**) of *S. paramamosain*. The scale of the y-axis is log10 in C. The expression pattern of *SCY3* in hepatopancreas (**D**) after challenge with *V. alginolyticus* and (**E**) after challenge with *S. aureus*. The asterisks indicated a significant difference compared with the control group (* *p* < 0.05). (**F**) mRNA expression in the pre- and post-mating reproductive system of *SCY3* in female *S. paramamosain*. The asterisks indicated a significant difference compared with the pre-mating group (* *p* < 0.05). Hc: hemocytes; Es: eyestalk; Gi: gills; Hp: hepatopancreas; Mu: muscle; Ne: nerve; Ht: heart; St: stomach; Mg: midgut; N: spermatheca; OA: ovaries; BW: subcuticular epidermis; RD: reproductive tract; SP: spermatheca; AVD: anterior vas deferens; PED: posterior ejaculation ducts; T: testis; ED: ejaculation duct; SV: seminal vesicle; PVD: posterior vas deferens. Copies in y-axis is copies/µg RNA.

**Figure 3 ijms-24-05689-f003:**
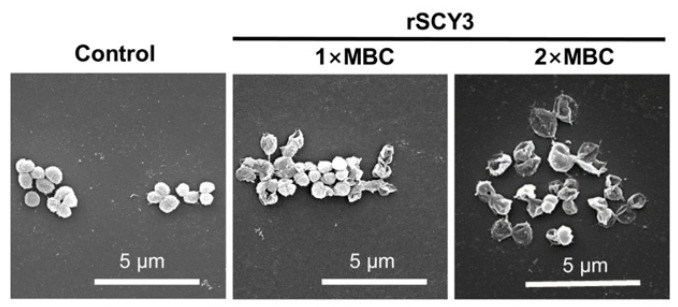
Effect of rSCY3 on morphology of *M. luteus*.

**Figure 4 ijms-24-05689-f004:**
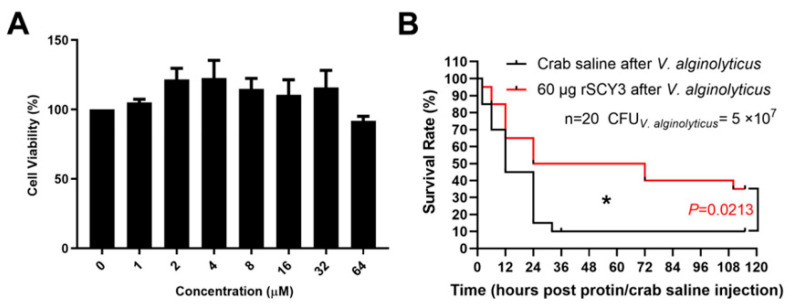
Evaluation of in vitro cytotoxicity and in vivo protective effect of rSCY3 on *S. paramamosain* infected by *V. alginolyticus*. (**A**) Cytotoxicity effect of recombinant protein rSCY3 on hemocytes isolated from male crabs. (**B**) In vivo protective effect of rSCY3. Crabs were challenged with *V. alginolyticus* at 5 × 10^7^ CFU, and injected with 60 μg rSCY3 at 1 h after bacterial challenging (n = 20 for each group). The survival curve of each group was analyzed using the Kaplan–Meier log-rank test. The asterisk indicated a significant difference compared with the control group (* *p* < 0.05).

**Figure 5 ijms-24-05689-f005:**
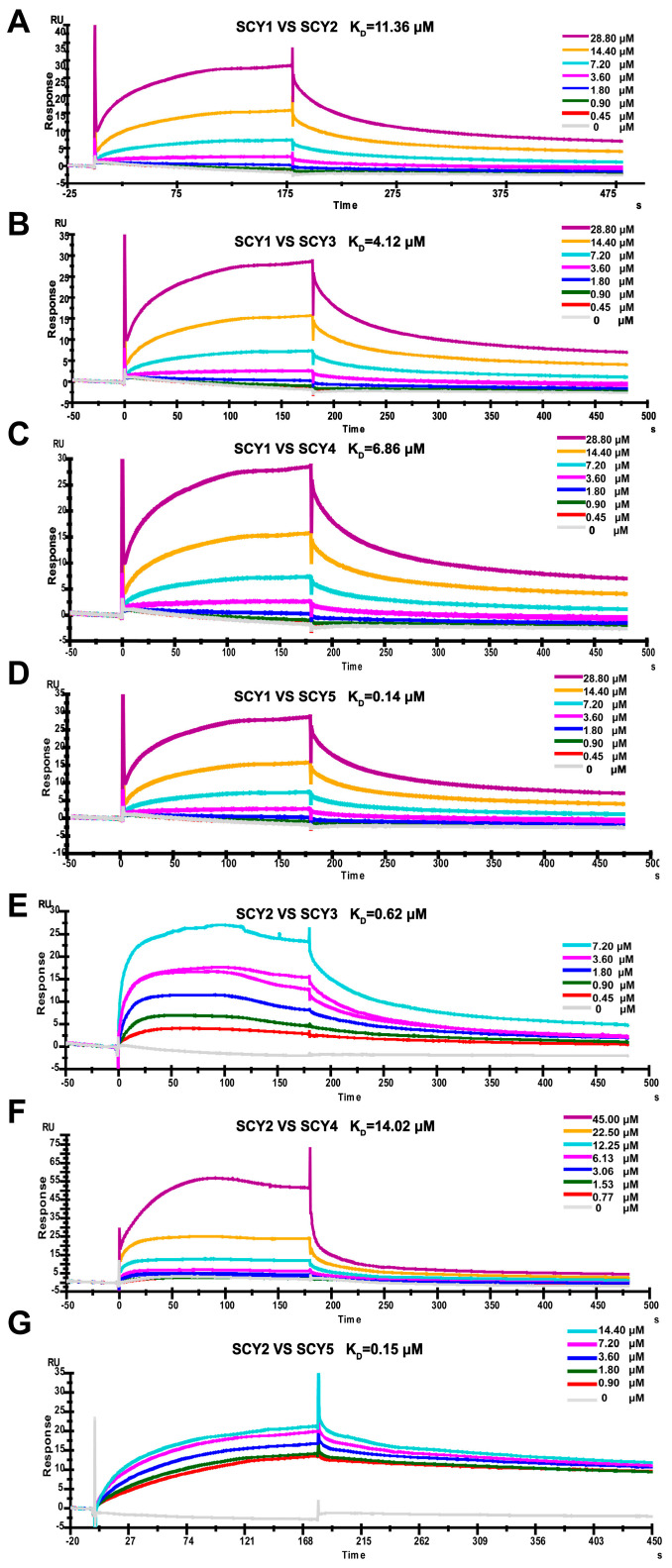
Binding kinetic between rSCY1 and rSCY2 (**A**); rSCY1 and rSCY3 (**B**); rSCY1 and rSCY4 (**C**); rSCY1 and rSCY5 (**D**); rSCY2 and rSCY3 (**E**); rSCY2 and rSCY4 (**F**); rSCY2 and rSCY5 (**G**), assessed by surface plasmon resonance (SPR, a technology for detecting interactions between biomolecules using biosensor chips).

**Figure 6 ijms-24-05689-f006:**
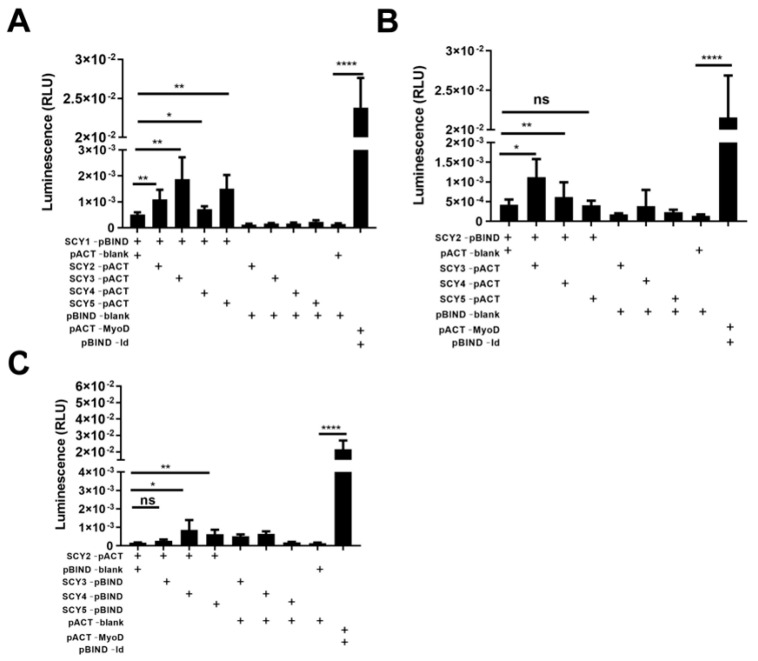
Verification of the rSCY1 and rSCY2, rSCY1 and rSCY3, rSCY1 and rSCY4, rSCY1 and rSCY5, rSCY2 and rSCY3, rSCY2 and rSCY4, and rSCY2 and rSCY5 interactions by mammalian two-hybrid (M2H) assay (a way of detecting interactions between proteins in vivo). (**A**) The luminescence of SCY1-pBIND and SCY2/SCY3/SCY4/SCY5-pACT after 24 h co-transfected in the HeLa cell. (**B**) The luminescence of SCY2-pBIND and SCY3/SCY4/SCY5-pACT after 24 h co-transfected in the HeLa cell. (**C**) The luminescence of SCY2-pACT and SCY3/SCY4/SCY5-pBIND after 24 h co-transfected in the HeLa cell. Data represent the means ± SDs from two independent experiments. * *p* < 0.05; ** *p* < 0.01; **** *p* < 0.0001; ns = no significant (one-way ANOVA, repeated measurement).

**Figure 7 ijms-24-05689-f007:**
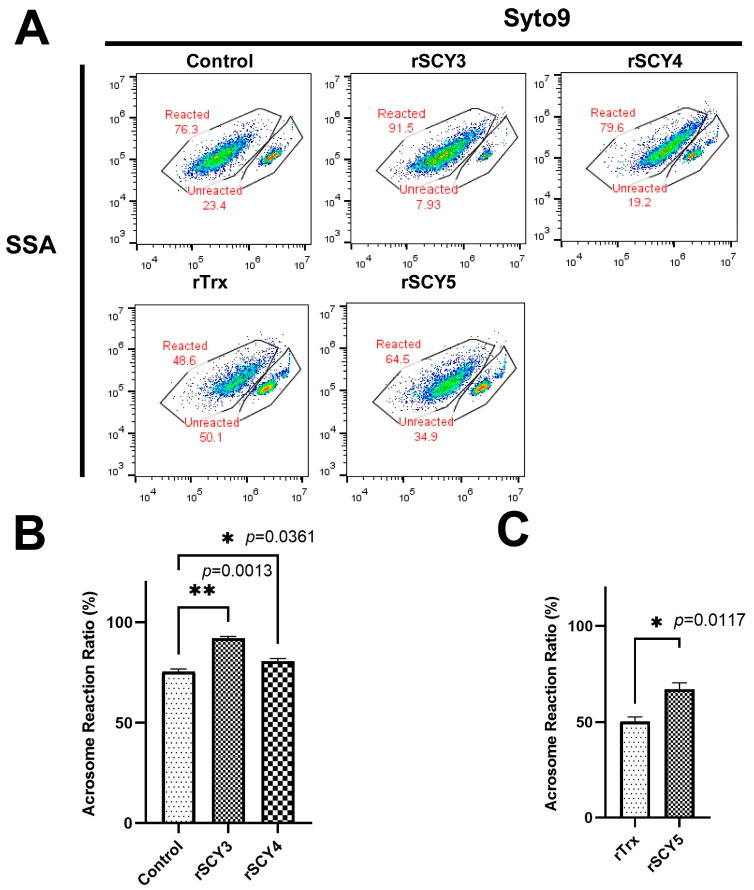
Flow cytometry analysis of the sperm % AR (**A**) and the quantification of changes after different treatments (**B**,**C**). The data shown are derived from a representative experiment reported as the mean (n = 2) ± SD. * *p* < 0.05; ** *p* < 0.01 (one-way ANOVA, repeated measurement).

**Figure 8 ijms-24-05689-f008:**
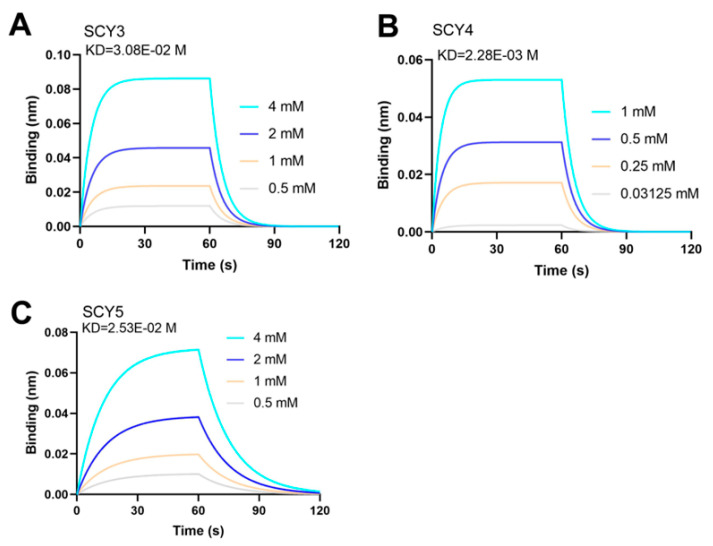
Affinity measurements between progesterone and rSCY3 (**A**); rSCY4 (**B**); and rSCY5 (**C**) by BLI.

**Table 1 ijms-24-05689-t001:** Antibacterial activity of rSCY3.

Microorganisms	CGMCC No. ^a^	*E. coli*- Derived SCY3		*P. pastoris*- Derived SCY3		rTrx	
		MIC ^b^	MBC ^b^	MIC	MBC	MIC	MBC
*Micrococcus luteus*	1.634	24–48	24–48	24–48	24–48	>48	>48

^a^ China general microbiological culture collection number; ^b^ MIC and ^b^ MBC: All the concentrations showed in this table were in μM. The values of MIC (minimal inhibitory concentration) and MBC (minimal bactericidal concentration) were expressed as the lowest concentration yielding no detectable microbial growth or that killed more than 99.9% of microorganism.

**Table 2 ijms-24-05689-t002:** Binding kinetics of rSCY1 and rSCY2, rSCY1 and rSCY3, rSCY1 and rSCY4, rSCY1 and rSCY5, rSCY2 and rSCY3, rSCY2 and rSCY4, and rSCY2 and rSCY5.

	K_D_ (µM)
rSCY1-rSCY2	11.26
rSCY1-SCY3	4.12
rSCY1-rSCY4	6.86
rSCY1-rSCY5	0.14
rSCY2-rSCY3	0.62
rSCY2-rSCY4	14.02
rSCY2-rSCY5	0.15

## Data Availability

Not applicable.

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
