# Peer review of "A New Gene SCY3 Homologous to Scygonadin Showing Antibacterial Activity and a Potential Role in the Sperm Acrosome Reaction of Scylla paramamosain"

_ijms, 2023, doi:10.3390/ijms24065689_

Round 1
Reviewer 1 Report
Introduction
Line 79. Please add a paragraph describing the major outcome obtained from SCY4 and SCY5 in Scylla paramamosain and then describe the reasons for studing SCY3 (this work).
Results
Line 154 delete “were analyzed”
L194 Replace “is deserved” by “deserves”
L 237-238. Please delete the sentence in this section or explain further in the discussion section: “these results further indicated that SCYs might act as important mediators of progesterone-induced AR”.
Discussion
L272. Delete that t. Add “T”
L292-293. Please, edit the paragraph “In the present study, the SCY3 showed similar characteristics of gene sequence and structure, we classified the known five SCYs as the SCYs AMP family. In its current way it is hard to follow.
Author Response
We thank the reviewer 1 for their valuable and insightful comments. We have sincerely considered the reviewer’s comments and tried our best to carefully revise our manuscript following reviewer’s comments. The responses to the reviewer’s comments item by item are in the files labeled "Response to Reviewer 1". Please see the attachment.
Thank you for receiving our manuscript and considering it for review. We appreciate your time and look forward to your response.

Reviewer 2 Report
The authors of this study are interested in the function of a new member of Scygonadin family of antimicrobial peptides in the mud crab, Scylla paramamosain. This new member, SCY3, is highly expressed in the ejaculatory ducts of the mud crab, and this expression is increased by challenge with V. alginolyticus and following mating in the female spermatheca. The authors nicely show the impact of this peptide on the morphology and culture of M. luteus (although it is not apparent why this species is used in this experiment), and the survival of infected mud crabs. Finally, the authors show that a number of these SCY peptides interact with each other and with progesterone, and have an impact on the acrosome reaction.
While this paper shows a lot of data, it is not always clear what the implications of this data are, or how it all fits together. The experiments are poorly explained – the results are stated, but little context is provided to explain the rationale for the experiment and the relevance of the results.
The following are several major recommendations:
1. As stated above, greater context and explanation would help the reader to follow the logic of the story presented.
2. While the data presented in Figures 1-4 flows logically, this clarity breaks down in Figures 5-7.
a. Figure 5 – the quality of this figure is too poor to evaluate, and there is minimal context provided with the results description. However, I wonder if this could be replaced by a simple table with Kd values. Unless there is additional information in this figure that I don't understand – in which case a better explanation would be beneficial. For example: why does it matter that these interact, and how does this lead directly to subsequent experiments? The clarity of the data would be much enhanced by an explanation of how these two methods (in figs 5 and 6) work.
b. Figure 6 – The axes are poorly labeled. Y-axis: a ratio involves two things. What are the two things? This is not clear to me, perhaps because the context for the experiment is not effectively explained. X-axis: The x-axes of these graphs are confusing. This could be improved by a table format for the axis, + and - indicating the presence or absence of the indicated items. Or perhaps a table with two rows: "DNA binding domain-" and "activation domain-"
c. Figure 7 – It appears that this figure should be presented as A-C (three panels). While I understand flow cytometry, it is not clear to me how this shows the acrosome reaction. In the figure I see a number of percentage values, but none match the percentages quoted in the text. This needs clarification.
3. The discussion needs significant rewriting. For example, it is not clear how the proteins described in lines 315-320 are relevant to the current study, except that they are found in the reproductive system (as are many other proteins). The reason for the factors described in many other sections of the discussion is likewise unclear.
The following are minor recommendations:
1. Figure 1: the color scheme in B should be explained in the legend. Extron should be exon.
2. Figure 2: the y-axis of A-C is absolute mRNA expression in copies. Is this copies per gram of tissue? Please clarify. The key for abbreviations should go at the end of the figure legend, since it applies to part F as well.
3. Figure 3. A scale bar should be found in each panel, or a statement made that the one applies to all. Above the figure it is suggested that Figure 3 shows that the cells died. I don’t think SEM can indicate that the cells died.
4. Table 1: there is no mention or explanation of rTrx as a control in the text.
5. Figure 4: A) Does an "increase" in cell viability indicate that rSCY3 stimulates growth? Some description of this and statistical analysis of any changes should be included. B) the y-axis is labeled ‘probability’. Is this real data or just a prediction of how crabs would die?
Author Response
We thank the reviewer 2 for their valuable and insightful comments. We have sincerely considered the reviewer’s comments and tried our best to carefully revise our manuscript following reviewer’s comments. The responses to the reviewer’s comments item by item are in the files labeled "Response to Reviewer 2". Please see the attachment.
Thank you for receiving our manuscript and considering it for review. We appreciate your time and look forward to your response.

Reviewer 3 Report
The manuscript ijms-2237831 entitled “A new gene SCY3 homologous to Scygonadin showing anti-bacterial activity and a potential role in the sperm acrosome re-action of Scylla paramamosain” provides comprehensive scientific results about the role of a new gene (SCY3) in the reproduction of Scylla paramamosain. In this study, the authors used several molecular techniques such as gene cloning, immune challenge experiments, antimicrobial assays, Scanning Electron Microscope, cytotoxicity assay, M2H assay, measuring the acrosome reaction, recombinant protein, etc. to reveal the function of SCY3. In brief, their results showed that SCY3 plays important roles in both immunity and reproduction (especially in the acrosome reaction) of S. paramamosain.
In general, the topic is important for the readers of IJMS and also for aquaculture scientists. The manuscript was well-written and well-structured. I would like to recommend the acceptance of this manuscript after some minor revisions and my specific comments are as below:
Abstract:
Line 17: What do you mean “transferred to the spermatheca of post-mating females at mating”? Do you mean this gene was in the spermatophores? Then the spermatophores transferred to females?
Line 19: “Staphylococcus aureus” must be italicized.
1. Introduction
Line 35-37: “Acrosome reaction” also has been studied in crustaceans. You can consider having a look at the below review paper: “Farhadi, A. and HarlıoÄŸlu, A.G., 2019. Molecular and cellular biology of the crayfish spermatozoon: toward development of artificial reproduction in aquaculture. Reviews in Fisheries Science & Aquaculture, 27(2), pp.198-214.”
Line 37-38: What kinda molecules are they? Are they genes? or proteins?
Line 38: Replace “et. al.,” with “, etc.”
2. Results
Line 95: What is “of”? I think you must remove “of” or replace it with “(ORF)”.
Line 177: “S. paramamosain” must be italicized.
Line 186: “S. par amamosain”. Remove the space between “r” and “a” in “S. paramamosain”.
Line 232: Do you have some photos from the AR in different treatments? If you have you can add it as a supplementary file.
Line 248: “3.08×10-2 M, 2.28×10-3 M and 2.53×10-2 M”. I think it should be written as “3.08×10-2 M, 2.28×10-3 M and 2.53×10-2 M. Check it carefully.
3. Discussion
Line 263: Remove “, one of the typical marine animals”
Line 263: [12,15,25-30,32-36]. Why a lot of references for this sentence? Is it necessary?
Line 271: maybe better to add “with” after “S. paramamosain infected”
Line 272: Remove “that “. Also, make “The” in capital letter. “The results....”
Line 275: Replace “effect on sperm“ with “affect the sperm”
Line 283: “Crustin”. Are you sure “C” must be capitalized?
Line 299: Did you use the tissue from “spermatheca in female” for qRT-PCR? Or you investigated the expression of the gene in the spermatophores/sperm that were collected from the female spermatheca? Please make this clear. Why say that “transferred to the spermatheca of female”. Maybe it is better to say “the spermatophores in the female spermatheca. Because the genes can not be transferred from the spermatophores into the tissue of the spermatheca in females.
Line 299: Do you think this gene is important for sperm capacitation? sperm capacitation in the female spermatheca is very important for AR and sperm fertility in crustaceans. For more information, you can read the below review paper: “Farhadi, A. and HarlıoÄŸlu, A.G., 2019. Molecular and cellular biology of the crayfish spermatozoon: toward development of artificial reproduction in aquaculture. Reviews in Fisheries Science & Aquaculture, 27(2), pp.198-214.”
Line 303: replace “In the” with “In this”
Line 322: “mud crab”
Line 334: replace “,” with a full stop “.”.
Line 337: replace “;” with a full stop “.”.
Line 338: replace “,” with a full stop “.”.
4. Materials and Methods
The word “previously described” or “described previously” is used about 10 times in the “Materials and Methods”.
Line 356: How many crabs?
Line 367: Did you publish those transcriptome databases? If yes cited. If you submitted the data to the NCBI please provide the accession number.
Line 376: “(Long, et al., 2021).” Must be cited according to the journal format.
Line 391-392: In several parts in the “Materials and Methods”, the authors cited a lot of papers for some quite simple experiments. For example, the collection of tissues or hemolymph is not that much complicated.
Line 451: “week”
Line 455: Give more information about the “crab saline” for example % of sodium chloride, etc.
Line 492: Replace “Sperm” with “Spermatophores”
Line 492: How did you extract the sperm from spermatophores?
Line 523: You said that “mean ± standard deviation” but later in “Line 230: means ± SEMs”. So It is confusing which one did you use for each date. In whole the manuscript you must use only one of them. “standard deviation” or “standard error”??
References:
“Wang, K.J” is cited about 15 times in this manuscript. The cited papers are quite related to this research. However, the authors might consider reducing the number of self-citations.
Author Response
We thank the reviewer 3 for their valuable and insightful comments. We have sincerely considered the reviewer’s comments and tried our best to carefully revise our manuscript following reviewer’s comments. The responses to the reviewer’s comments item by item are in the files labeled "Response to Reviewer 3". Please see the attachment.
Thank you for receiving our manuscript and considering it for review. We appreciate your time and look forward to your response.

Round 2
Reviewer 2 Report
This paper is much improved. Several errors are still noted by this reviewer.
· Figure 1 figure legend: replace ‘consistency’ with ‘conservation’
· Figure 2C: the y-axis should be a logarithmic scale – this would be better than the breaks in the current y-axis.
· In Response 1-a, the authors state they investigated the antimicrobial effect of SCY3 on 23 bacteria and 8 fungi. I don’t see this data in the paper, or in the supplement. Unless I am missing this, I think it should be included. This negative data makes the effect on M. luteus even more significant.
Author Response
Thanks for the reviewer’s careful review and professional advice. We have revised our manuscript following the reviewer’s comments carefully. The revised parts are uploaded with revision trace in the resubmitted manuscript. And responses to the reviewer’s comments point to point are also submitted in the files labeled "Response to Reviewer 2". Please see the attachment.
